# Simultaneous polyclonal antibody sequencing and epitope mapping by cryo electron microscopy and mass spectrometry

Douwe Schulte[1], Marta Šiborová[1], Lukas Käll[2], Joost Snijder[1]*

[1]Biomolecular Mass Spectrometry and Proteomics, Bijvoet Center for Biomolecular Research and Utrecht Institute of Pharmaceutical Sciences, Utrecht University, Padualaan, Utrecht, Netherlands; [2]Science for Life Laboratory, School of Engineering Sciences in Chemistry, Biotechnology and Health, Royal Institute of Technology – KTH, Solna, Sweden

*For correspondence:
j.snijder@uu.nl

Competing interest: The authors declare that no competing interests exist.

## eLife Assessment

The paper addresses the problem of optimising the mapping of serum antibody responses against a known antigen. The manuscript describes a method using EM polyclonal epitope mapping to help elucidate endogenous antibodies. The work is interesting and **valuable** to the fields of immunology and serology, and the strength of evidence to support its findings is considered **solid**.

**Abstract** Antibodies are a major component of adaptive immunity against invading pathogens. Here, we explore possibilities for an analytical approach to characterize the antigen-specific antibody repertoire directly from the secreted proteins in convalescent serum. This approach aims to perform simultaneous antibody sequencing and epitope mapping using a combination of single particle cryo-electron microscopy (cryoEM) and bottom-up proteomics techniques based on mass spectrometry (LC-MS/MS). We evaluate the performance of the deep-learning tool ModelAngelo in determining de novo antibody sequences directly from reconstructed 3D volumes of antibody-antigen complexes. We demonstrate that while map quality is a critical bottleneck, it is possible to sequence antibody variable domains from cryoEM reconstructions with accuracies of up to 80–90%. While the rate of errors exceeds the typical levels of somatic hypermutation, we show that the ModelAngelo-derived sequences can be used to assign the used V-genes. This provides a functional guide to assemble de novo peptides from LC-MS/MS data more accurately and improves the tolerance to a background of polyclonal antibody sequences. Following this proof-of-principle, we discuss the feasibility and future directions of this approach to characterize antigen-specific antibody repertoires.

## Introduction

Adaptive immunity to invading pathogens is mediated to an important degree by antibodies (*Bonilla and Oettgen, 2010*; *Rees, 2020*; *Burton, 2023*; *Lam et al., 2020*). The available repertoire of antibodies is unique in individuals and constantly shifting in response to immunological pressure under which activation, selection, proliferation, maturation, and differentiation of the antibody-producing B-cells takes place (*Lam et al., 2020*; *Tellier and Nutt, 2019*; *Pieper et al., 2013*). Therefore, understanding the molecular mechanisms of antibody-mediated immunity requires knowledge of how a

complex repertoire of polyclonal antibodies targets a diverse landscape of epitopes on their respective antigens.

The analytical challenge at hand is to resolve antigen-antibody interactions down to the pairwise contacts between specific amino acid residues in the epitope and paratope regions, respectively. This would enable reconstruction of the evolutionary pathways of somatic recombination and hypermutation that lead to high-affinity antigen binding. Conversely, this would also reveal how this selective immune pressure drives the evolutionary pathways of antigenic drift in targeted pathogens (*Marks and Deane, 2020*; *Han et al., 2023*; *Vajda et al., 2021*; *White, 2021*). Knowledge of the precise antibody sequences, as well as near-atomic details of the epitope-paratope interaction, are thus prerequisites to understand the coevolution between replicating pathogens and antibody-mediated adaptive immunity in the host.

Current methods to determine the antigen-specific antibody repertoire rely on single (memory) B-cell sorting, followed by targeted sequencing of the coding mRNAs for the heavy and light chains (*Georgiou et al., 2014*; *Lavinder et al., 2015*). This enables the production of recombinant monoclonal antibodies, whose epitopes may be mapped to near-atomic structural detail by X-ray crystallography and cryo electron microscopy (cryoEM). This approach has generated a wealth of information about antibody-antigen interactions, though it is biased by the limited pool of memory B-cells that it probes. Antibodies function as circulating glycoproteins in bodily fluid, secreted from plasma cells which are in turn derived from diverse pools of memory B cells located in various tissues, including bone marrow, spleen, lymph nodes, and only to a minor degree in blood (*Inoue and Kurosaki, 2024*; *Meng et al., 2017*; *Akkaya et al., 2020*). Serological assays aimed to determine binding and neutralization titers specifically look at the secreted antibody in bodily fluid, and it remains an outstanding question how this 'serum compartment' of the antibody repertoire relates both qualitatively and quantitatively to the minor population of memory B-cells found in peripheral blood. This calls for new analytical approaches that can derive both antibody sequence and epitope information straight from the secreted antibody product.

Such approaches have been developed in recent years, based on mass spectrometry and electron microscopy. First, using a bottom-up proteomics approach, antibody-derived peptides can be sequenced de novo from fragmentation spectra and assembled into full heavy/light chain sequences (*Lavinder et al., 2015*; *de Graaf et al., 2022*). Sequence accuracy is such that functional monoclonal antibodies can be reconstructed from the input data, and several reports have described successful sequencing efforts of human serum, milk, and urine-derived antibodies (*Fridy et al., 2014*; *Peng et al., 2023b*; *Bondt et al., 2024*; *Peng et al., 2023a*; *Tran et al., 2016*; *Bondt et al., 2021*; *Sousa et al., 2012*; *Sen et al., 2017*; *Rickert et al., 2016*; *Savidor et al., 2017*; *Peng et al., 2021*; *Guthals et al., 2017*; *Cheung et al., 2012*; *Castellana et al., 2011*; *Bandeira et al., 2008*; *Dupré et al., 2021*; *Peng et al., 2025*). Second, both hydrogen-deuterium exchange mass spectrometry and electron microscopy have been used to resolve a complex landscape of epitopes targeted by polyclonal antibody mixtures (*Antanasijevic et al., 2022a*; *Antanasijevic et al., 2022b*; *Bangaru et al., 2022*; *Boyoglu-Barnum et al., 2021*; *Dingens et al., 2021*; *Grauslund et al., 2024*; *Han et al., 2021*; *Nogal et al., 2020*; *Ständer et al., 2021*; *Zhang et al., 2013*; *Vorauer et al., 2024*). Ward and colleagues have reported that with the latter approach, which they coined Electron Microscopy based Polyclonal Epitope Mapping (EMPEM), they obtained near-atomic resolution reconstructions by cryoEM to resolve side-chain densities in the epitope-paratope region (*Antanasijevic et al., 2022a*). This opens the possibility to derive antibody sequence information and integrate this into the structural modelling of the interaction. In essence, the reconstructions might reveal which antibodies from the complex polyclonal mixture bind which epitopes on the antigens. The improved resolutions of current cryoEM approaches thus allow for a type of visual proteomics in which protein identity (i.e. antibody sequence) may be directly inferred from the reconstructed 3D volumes (*Robinson et al., 2007*; *Leung et al., 2023*; *Gui et al., 2022*; *Gui et al., 2021*; *Schmidt et al., 2024*; *Fianu et al., 2024*; *Cingolani et al., 2024*; *Jiang et al., 2022*; *Hugener et al., 2024*). While the pure sequencing accuracies from these approaches is obviously limited by resolution/map quality, many tools have been recently developed to infer protein identity in an automated fashion, including cryoID, DeepTracer-ID, *findmysequence*, and most recently ModelAngelo (*Ho et al., 2020*; *Chang et al., 2022*; *Chojnowski et al., 2022*; *Jamali et al., 2024*).

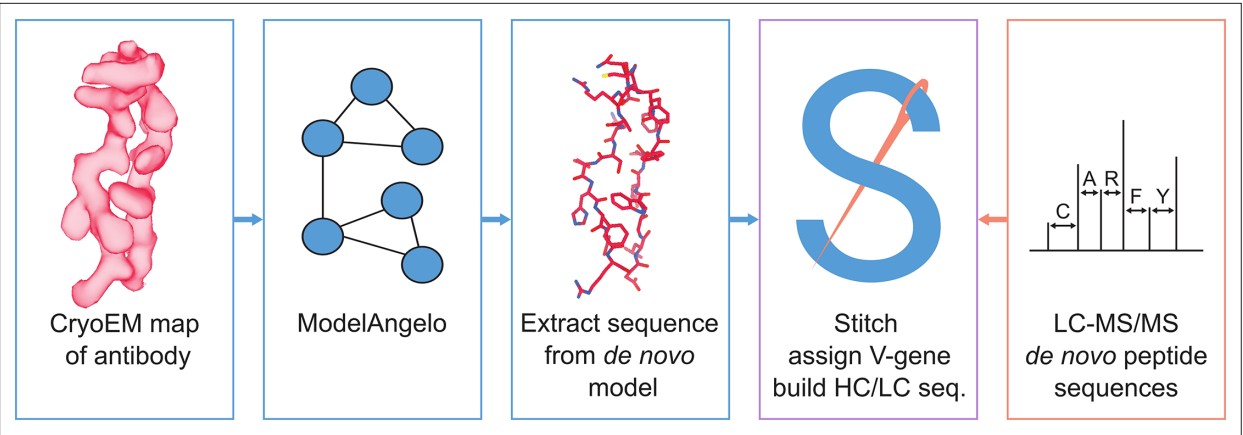

**Figure 1.** Schematic workflow to estimate de novo antibody sequences by the integration of cryoEM and LC-MS/MS data with Stitch.

Here, we explore the use of ModelAngelo to derive de novo antibody sequences from experimental cryoEM density maps of antibody-antigen complexes. We have previously developed the software tool Stitch, which sorts and assembles MS-derived peptide sequences into full heavy/light chain sequences across complex repertoires (*Schulte et al., 2022*; *Schulte and Snijder, 2024*). We adapted Stitch to perform the same task on ModelAngelo-derived de novo models and test the accuracy of the approach on a benchmark of 164 publicly available cryoEM maps of monoclonal antibody-antigen pairs. We demonstrate that map quality is a critical bottleneck, but that antibody sequences can be derived with up to 80–90% accuracy. We test the utility of these sequences for assigning the used V-genes, which together with reconstruction of CDRH3 may offer a useful guide to assemble more accurate MS-derived peptide sequences (see *Figure 1*). We show that such EM-derived templates indeed improve MS-based sequencing accuracy in the context of complex antibody mixtures and that publicly available EMPEM reconstructions are of sufficient quality to leverage this approach. This proof-of-principle offers a promising perspective to integrate cryoEM and MS methods for a comprehensive characterization of the antibody repertoire on both sequence and epitope levels.

## Results

To assess the feasibility of deriving de novo antibody sequences from experimental cryoEM density maps, we assembled a benchmark dataset from the Electron Microscopy Data Bank (see *Supplementary file 1*). To infer the antibody sequences, we chose the recently published deep-learning tool ModelAngelo, developed by Jamali, Scheres, and colleagues, as it is capable of inferring complete de novo models in cryoEM density maps without the need for user input sequences or main chain models (*Ho et al., 2020*; *Chang et al., 2022*; *Chojnowski et al., 2022*; *Jamali et al., 2024*). We searched the EMDB for published maps containing antigen-binding fragments (Fabs) of any species, at nominal resolutions of ≤4 Å, released after the training data for ModelAngelo was obtained. These results were filtered for maps that included a deposited atomic model and which contained only a single unique monoclonal Fab (of which multiple copies may be bound in the reconstructed antigen complex). The final benchmark includes 164 maps, including Fabs from human, rabbit, mouse, and macaque species. The maps were used as input data for ModelAngelo without user-provided sequences, yielding completely de novo atomic models of both antigen and Fabs.

The output models from ModelAngelo are typically fragmented to varying degrees depending on the local quality of the map. In addition, maps may contain multiple copies of the same unique Fab molecule. We therefore aimed to consolidate all fragments for the built Fabs into a single consensus sequence for the antibody variable domains of the heavy and light chains. We have previously developed the software Stitch, which performs assembly of LC-MS/MS derived de novo peptide reads into the correct framework of the heavy and light chains by alignment to germline-template sequences from the ImMunoGeneTics (IMGT) database (*Manso et al., 2022*; *Lefranc, 2014*; *Lefranc et al., 2015*). We adapted Stitch to use models from ModelAngelo (or any mmCIF file) as input data, extract the amino acid sequences, and perform the same template-based assembly with the resulting reads.

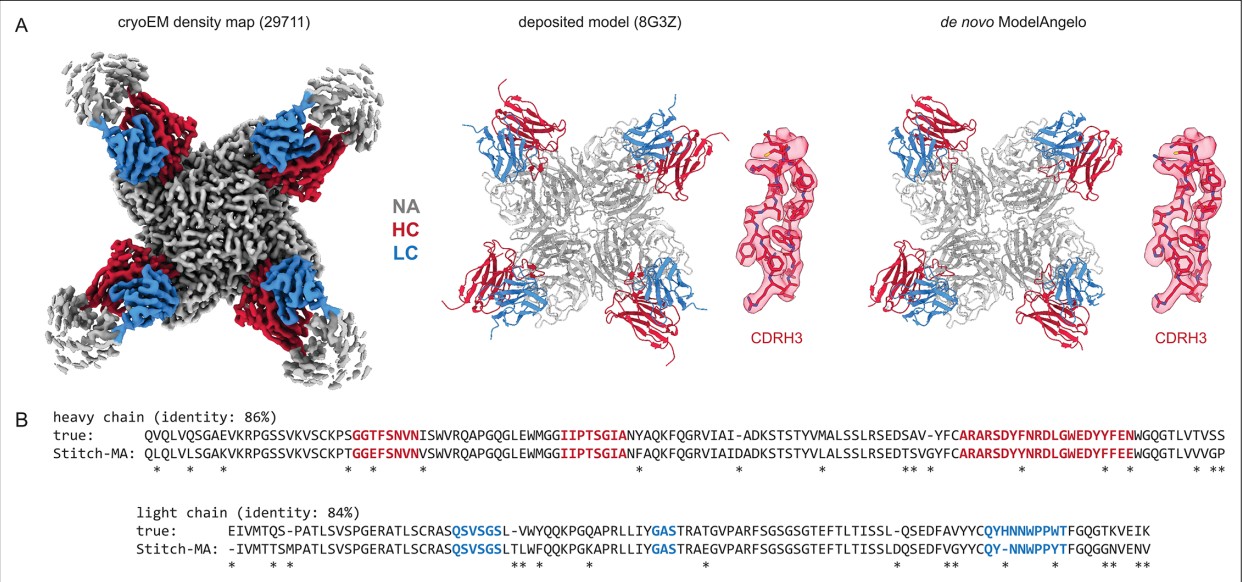

**Figure 2.** Determining de novo antibody sequences from cryoEM data with ModelAngelo. (**A**) Exemplary map (with top10% alignment scores of 1076/1174 for HC/LC) from the benchmark dataset, representing an Influenza B virus neuraminidase (NA) in complex with four copies of a neutralizing Fab, at global FSC resolution of 2.3 Å. Shown are the deposited map, model, and the de novo model generated by ModelAngelo, along with a detailed view of CDRH3. (**B**) Consensus sequences for heavy and light chains as generated by Stitch compared to the true sequences. Sequencing errors are indicated by an asterisk (*).

Of the 164 input maps, 141 and 144 yielded a non-zero alignment score for the heavy and light chains in the Stitch result, respectively. These add up to a total of 152 maps which were analyzed further to evaluate the sequence accuracy of this approach (including 134 maps with a non-zero alignment score for *both* heavy and light chain).

Results for one of the top-scoring entries in the benchmark, an Influenza B virus neuraminidase tetramer in complex with four identical copies of a neutralizing Fab (*Momont et al., 2023*), is shown in *Figure 2*. The consensus sequences generated in Stitch have complete coverage of both heavy and light chain variable domains, including both CDRL3 and CDRH3. The de novo determined sequence is 84% and 86% identical to the true heavy and light chain sequences, respectively. The 15% rate of sequencing errors exceeds the typical levels of somatic hypermutation observed in mature antibody sequences, which are on the order of 1–10%. While the derived sequence should therefore not be taken at face value to reconstruct recombinant monoclonal antibodies, we reasoned that the accuracy

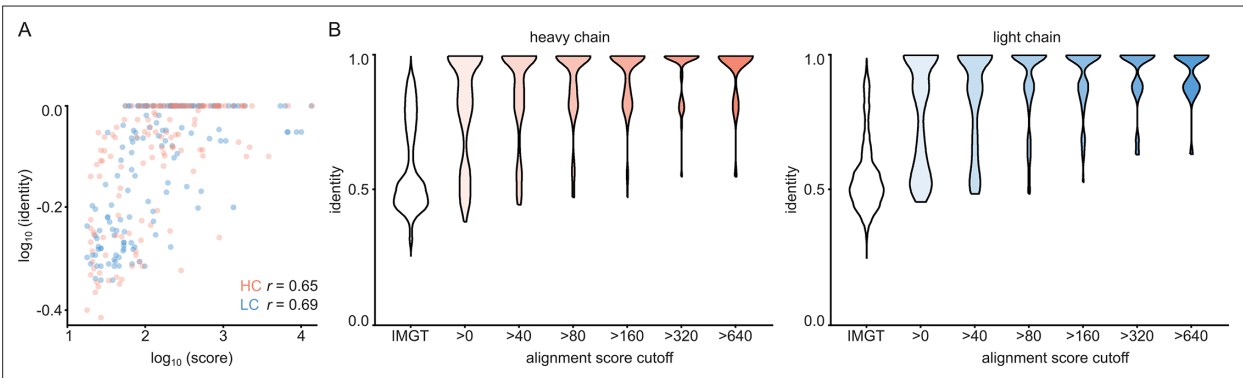

**Figure 3.** V-gene assignment from ModelAngelo data. (**A**) Correlation between Stitch alignment score and sequence identity between the top-scoring V-gene of the ModelAngelo vs PDB sequence of the heavy and light chain variable domains, as indicated with the non-parametric Spearman correlation coefficient. (**B**) Distribution of V-gene sequence identity for progressive alignment score cutoffs, compared to the pairwise V-gene sequence identity in the IMGT repertoire.

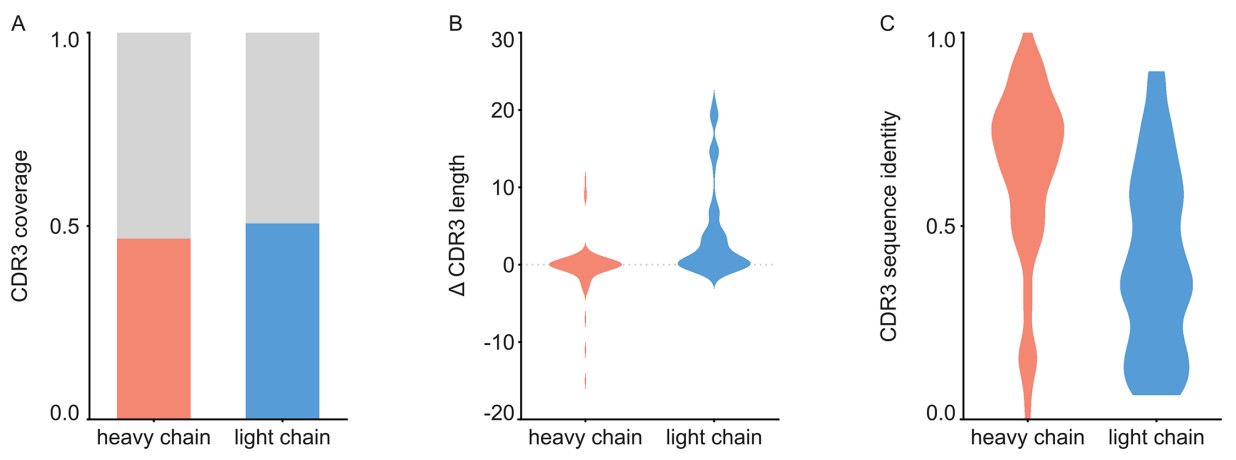

**Figure 4.** Analysis of de novo CDR3 modeling in ModelAngelo-Stitch. (**A**) Coverage of CDR3 for the heavy and light chain. CDR3 was counted for coverage if the de novo sequence spanned the flanking cysteine on the V gene and the tryptophan or phenylalanine on the J gene. Proportion of maps with CDR3 coverage in red/blue, maps with missing CDR3 in grey. (**B**) Difference in length between de novo modelled CDR3 vs. true sequence. (**C**) Sequence identity of de novo modelled CDR3 vs. true sequence.

is nevertheless likely sufficient to correctly infer the corresponding germline V-genes of the mature antibodies.

For each of the 152 maps, we compared the pairwise sequence identity between the top scoring V-genes from the ModelAngelo input in Stitch with the top scoring V-genes from the true sequences (as deposited in the corresponding PDB entry). For reference, we also calculated the pairwise sequence identities of all available V-gene templates per species, reflecting what a completely random draw from the available V-gene repertoire would look like. As shown in *Figure 3*, the top scoring V-genes from the ModelAngelo sequences have significantly higher pairwise identities than a random draw from the V-gene repertoire in IMGT for both the heavy and light chains (p<0.0001 in unpaired, two-tailed Kolmogorov Smirnov tests). Furthermore, the identity of the inferred V-gene with the true sequence scales with the alignment scores in Stitch, making it a valuable metric for the quality of the V-gene inference. The mean/median V-gene identity in a random draw from the IMGT repertoire is 0.59/0.52 and 0.56/0.53 for the heavy and light chains, respectively. With the ModelAngelo-derived sequences this improves to 0.78/0.87 and 0.75/0.85 for the heavy and light chains in the full dataset.

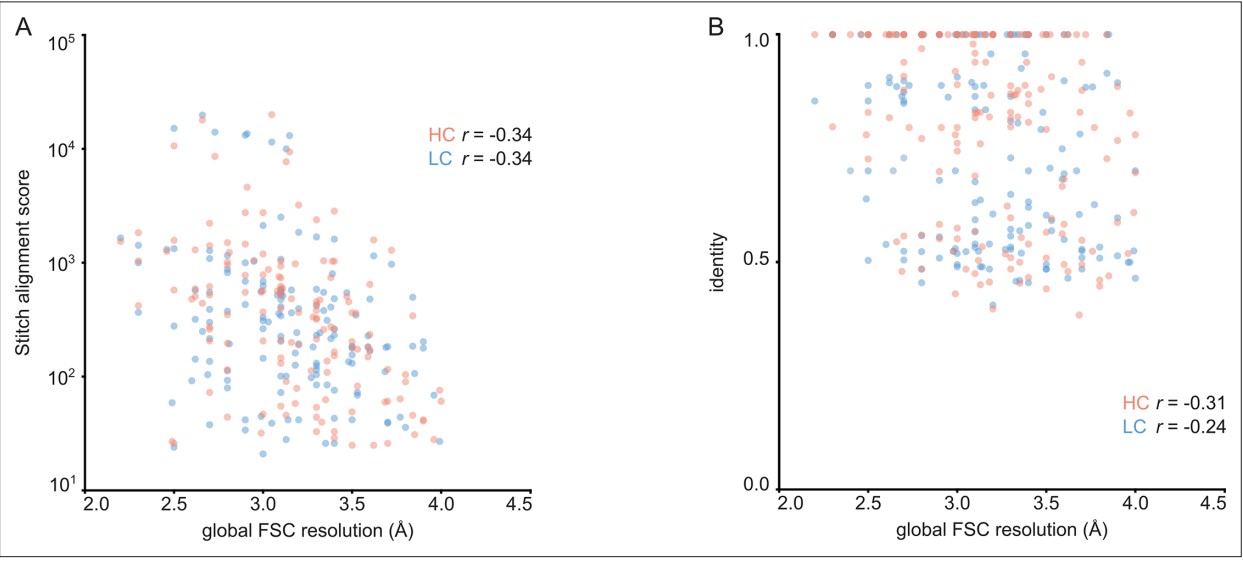

**Figure 5.** Correlation between global FSC resolution and Stitch alignment score (**A**) or inferred V-gene identity (**B**). The non-parametric Spearman correlation coefficient is indicated for heavy and light chain.

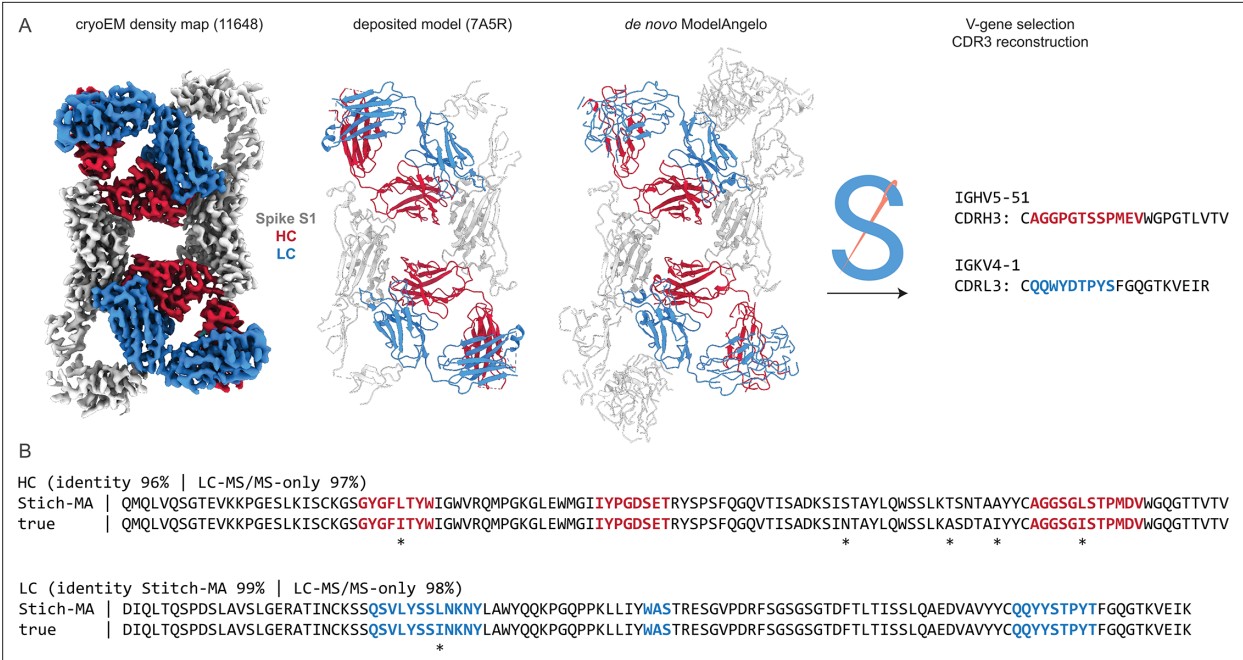

**Figure 6.** Sequencing CR3022 with integrated cryoEM and LC-MS/MS data. (**A**) Shown are the deposited cryoEM map (global FSC resolution 4.1 Å), model, and de novo ModelAngelo output for the CR3022 Fab in complex with the SARS-CoV-2 Spike S1 subunit. The sequences were extracted from the de novo model and used as input for Stitch, resulting in the identification of the indicated V-genes and CDR3 sequences. These variable domains were used as templates in Stitch to assemble the LC-MS/MS derived de novo peptides. (**B**) Consensus sequences for CR3022 from the integrated cryoEM-MS data in Stitch compared to the true heavy and light chain sequences. Sequencing errors are indicated with an asterisk (*) .

This gradually improves with higher alignment scores to 0.89/1.00 and 0.88/1.00 for heavy and light chains, starting at a cutoff of 80 (representing 92/141 and 80/134 maps for heavy and light chain, respectively). The complete complementarity determining regions CDRH3 and CDRL3 were covered in 66 and 68 maps for the heavy and light chain, respectively, see *Figure 4*. The length of the complete antigen binding loops was estimated with an average error of 0.5±3.3 or 1.7±6.0 residues for heavy and light chain, with average sequence identities of 0.63 and 0.41. While CDRH3 is the more challenging region in MS-based approaches to antibody sequencing, we believe that the moderately better length and sequence accuracy of CDRH3 compared to CDRL3 in ModelAngelo output reflects the CDRH3's notoriously tight involvement in antigen binding, hence a greater relative stability in the antibody-antigen complex, resulting in better order in the reconstructed EM density maps. We found the global FSC resolution of the input map to be a poor predictor of both the Stitch alignment score and the inferred V-gene identity, likely because it is dominated by the bulk of the antigen and not representative of the local resolution in the epitope-paratope region as shown in *Figure 5*. These results demonstrate that candidate V-genes for the antibodies resolved in cryoEM densities can be accurately narrowed down using ModelAngelo and Stitch and that a limited subset of maps contains accurate information on CDR3 sequence and length.

In the context of polyclonal antibody mixtures, this analysis suggests that cryoEM densities of antigen-antibody complexes from EMPEM experiments can be leveraged to guide sequence assembly from complementary proteomics-based profiling of the same sample (see *Figure 1*). In such an experiment, reconstructed cryoEM densities would be used as input data for ModelAngelo, from which the sequences are extracted and run through Stitch to select the top-scoring V-gene and construct a placeholder sequence for CDR3 of both the heavy and light chain. These reconstructed variable domains may then act as templates to guide the assembly of de novo peptides from LC-MS/MS data to improve the accuracy of the candidate sequence.

As a proof-of-principle, we tested this on the monoclonal antibody CR3022, for which both a cryoEM reconstruction and LC-MS/MS data are publicly available (*Figure 6*). This antibody was isolated from a convalescent survivor of a SARS-CoV infection and targets a cryptic, but conserved epitope at the base of the Spike receptor-binding domain, with cross-neutralization to SARS-CoV-2

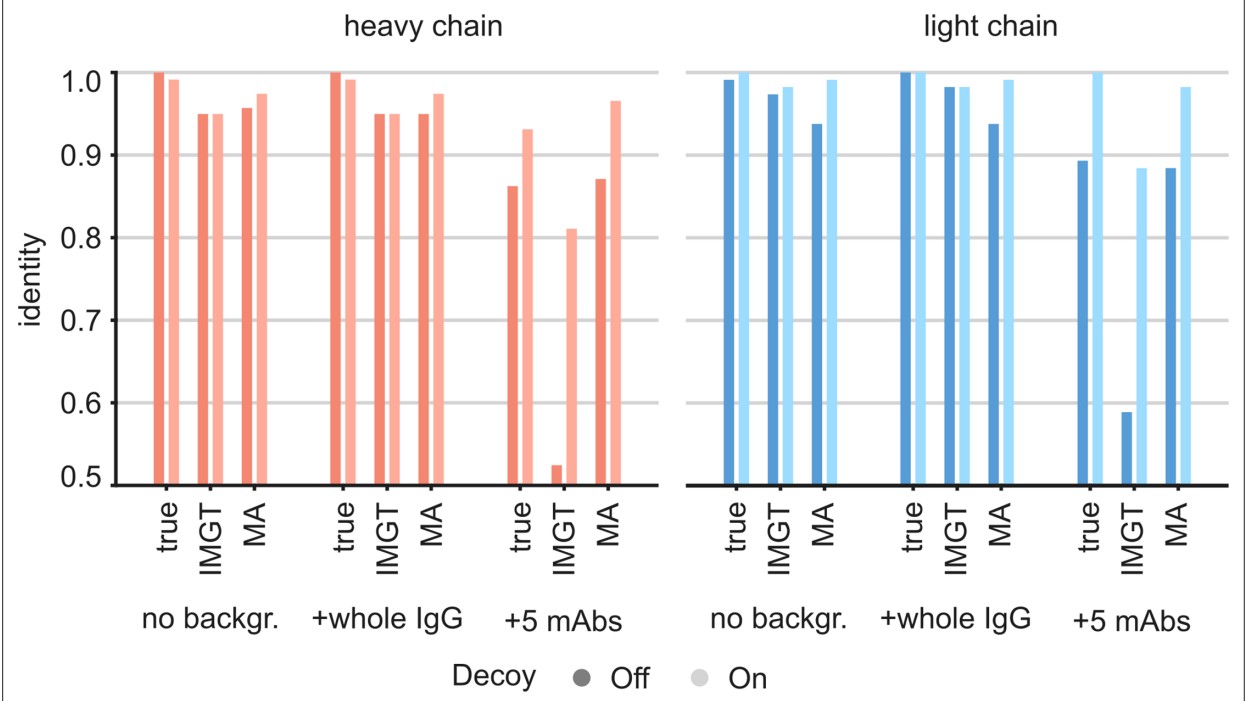

**Figure 7.** Targeted sequencing of CR3022 against a complex background of other antibodies. Plotted are the de novo consensus sequence identities derived from the LC-MS/MS data using either the true sequences, the full IMGT repertoire, or the ModelAngelo-derived variable domains as templates. We compare the output from the CR3022 dataset alone ('No backgr.') with the output after adding either a diffuse polyclonal IgG background from a COVID-19 patient ('+whole IgG') or full datasets from five additional anti-Influenza-HA monoclonals ('+5 mAbs'). Use of decoy sequences as indicated by dark/light colors.

(*ter Meulen et al., 2006*; *Yuan et al., 2020*). The antibody consists of an IGHV5-51 heavy chain, paired with an IGKV4-1 light chain. When complexed with full-length SARS-CoV-2 Spike, its Fab induces an odd rearrangement of the Spike protomers to yield an antiparallel dimer of S1 subunits in the cryoEM reconstructions (*Wrobel et al., 2020*; *Huo et al., 2020*). When using this map as input for ModelAngelo and subsequently Stitch, the IGHV5-51 and IGKV4-1 germline sequences are correctly identified based on their alignment scores. Moreover, complete sequences for CDR3 of the heavy and light chains are built. When using these reconstructed variable domains as templates to guide assembly of the de novo peptide reads from the LC-MS/MS data published by Person and colleagues (*Gadush et al., 2022*), the final consensus sequences are 96% and 99% identical to the true heavy and light chain respectively. Of note, the only three remaining errors in the six CDR sequences are I/L assignments, which have identical masses and are notoriously challenging for MS-based sequencing.

For this CR3022 dataset, derived from the monoclonal antibody, the LC-MS/MS data alone assembled with Stitch against the full range of V-genes already yields a similar accuracy of 97% and 98% for the heavy and light chain, respectively. For the case of an EMPEM experiment, the challenge would rather be to correctly assemble the CR3022 sequence against a background of unrelated antibody sequences in a complex mixture. We therefore tested the utility of the ModelAngelo-derived templates for sequence assembly by also mixing input reads from LC-MS/MS data of unrelated antibodies (*Figure 7*). First, we mixed in a background of whole IgG from a hospitalized COVID-19 patient (*Schulte et al., 2022*), representing a diffuse polyclonal background in approximately a 1:1 ratio to the target input reads. Second, we mixed in a background of five additional unrelated anti-Influenza-HA monoclonal antibodies from the same study of Person and colleagues (*Gadush et al., 2022*), amounting to a 5:1 ratio of background to target input reads. In our previous work on sequencing serum-derived antibodies by bottom-up proteomics with Stitch, we found that assembly of the consensus sequences becomes much more tolerant to background data if, beyond the top scoring V-gene, the remaining unrelated V-genes are also included as decoys for the final template matching step (*Schulte et al., 2022*). The use of these decoy sequences is also included in the comparison here. Furthermore, we

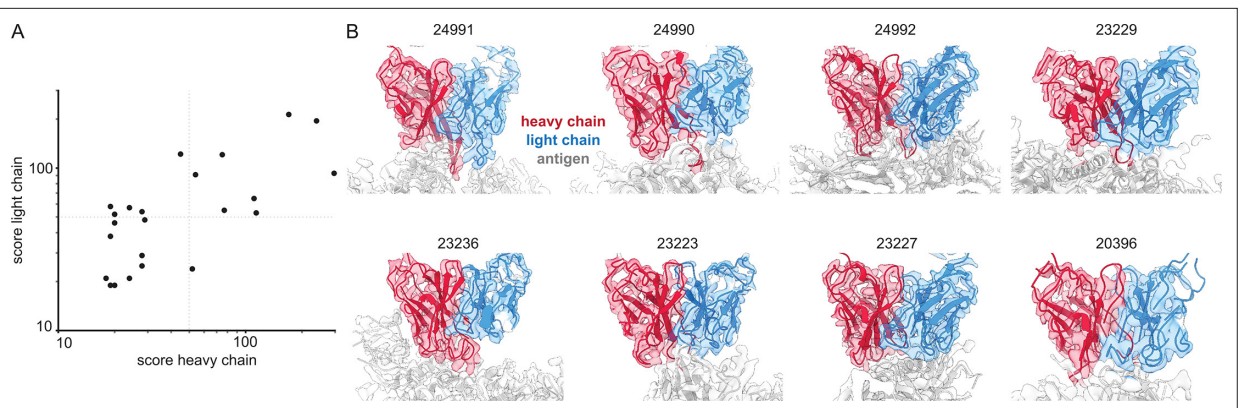

**Figure 8.** Inferring V-genes from published EMPEM data. (**A**) Plotted are all non-zero alignment scores in Stitch from published EMPEM maps. (**B**) Views of the variable domains of EMPEM maps with alignment scores >50 for both heavy and light chains. The EMDB identifiers are indicated at each panel.

also included the use of the true CR3022 sequences as templates, serving as a best-case scenario, positive control. The analysis confirmed that even without ModelAngelo-derived templates, the sequence assembly with Stitch is already tolerant to the diffuse polyclonal background from the whole IgG fraction, yielding a similar accuracy as in the absence of these background peptides. By contrast, the sequence accuracy plummets to below 0.6 when the background consists of the five additional monoclonal antibodies in a 5:1 ratio to the target input reads. The accuracy is recovered to >0.95 when using either the true CR3022 sequences or the ModelAngelo-derived templates. There is also a gain in accuracy by using decoy templates with the LC-MS/MS data alone, although smaller compared to the use of ModelAngelo-derived templates. These results demonstrate that ModelAngelo-derived templates are useful for sequence assembly against complex polyclonal backgrounds.

We have demonstrated that cryoEM reconstructions of monoclonal antigen-antibody complexes may contain sufficient information to accurately narrow down candidate V-genes and that this can be integrated with proteomics data to improve the accuracy of candidate sequences. We also evaluated whether EMPEM data is indeed of sufficient quality to infer V-genes from automated de novo modeling in the maps (*Figure 8*). We downloaded all 46 published EMPEM maps from EMDB, of which 23 were of sufficient quality to give a non-zero alignment score when using the ModelAngelo results in Stitch (see *Supplementary file 2*). Analysis of the benchmark of monoclonal antigen-antibody complexes showed that the quality of the V-gene inference scales with the alignment scores. In this set of EMPEM reconstructions, the alignment scores range from ca. 20–300. Of the 23 maps, 8 have alignment scores above 50 for both heavy and light chain, at which point we estimate the mean/median V-gene identity to be approximately 0.85/0.90. This analysis shows that experimental EMPEM studies may yield sufficiently detailed reconstructions of the antigen-antibody complexes to narrow down the candidate V-genes of the resolved Fabs accurately.

## Discussion

The development of EMPEM and MS-based polyclonal antibody sequencing now make it possible to profile the antigen-specific antibody repertoire straight from the secreted pool of antibodies in bodily fluids. This approach bridges the gap between established single B-cell sequencing approaches and serological assays to probe binding and neutralization titers. While sample complexity remains an important bottleneck, and questions remain about the dynamic range of the true serum antibody repertoire and the depth of coverage from these novel experimental approaches, several studies have recently reached the important milestone of reconstructing functional antibodies from direct measurements of the secreted serum components (*Fridy et al., 2014*; *Bondt et al., 2024*; *Peng et al., 2023a*; *Bondt et al., 2021*; *Sousa et al., 2012*; *Rickert et al., 2016*; *Guthals et al., 2017*; *Castellana et al., 2011*; *Antanasijevic et al., 2022a*; *Ferguson et al., 2024*). The present work demonstrates that epitope and sequence information can be integrated using ModelAngelo and Stitch. This approach holds promise to better understand the serum compartment of the antibody repertoire. Use of the cryoEM data in these workflows complements the MS data beyond epitope mapping in several

significant ways. First, we demonstrated that narrowing down the candidate V-genes improves the tolerance of LC-MS/MS-derived peptide sequence assembly to background in a complex antibody mixture. Second, heavy-light chain pairing is a problematic blind spot to proteomics sequencing, as the antibodies are denatured and digested as part of the sample workup. In contrast, this pairing is trivial in cryoEM data as the chains are in direct contact in the resolved Fabs in the map. Finally, reconstruction of CDRH3 is especially challenging with proteomics data alone, as it spans the junction between the recombined V-, D-, and J-segments, of which the D-segment is short and hypervariable to a point that germline sequences do not provide a functional template for sequence assembly. While CDRH3 coverage is also limited in ModelAngelo data, it can be built in many cases and a correct estimate of CDRH3 length is already useful to guide assembly of the de novo peptide reads.

Here, we have extracted the flat sequences from ModelAngelo output to use as input for the template matching step in Stitch, using a modified Smith-Waterman Alignment. We believe the template matching step in Stitch could be further improved in several ways. First, ModelAngelo has a built-in database search based on *HMMer* using profile Hidden Markov Models (HMM) encoding the amino acid probabilities across the alphabet at each position (*Mistry et al., 2013*). While the ModelAngelo search currently does not consolidate the fragmented output models into a single search to build consensus sequences, we anticipate that implementing a similar HMM profile search in Stitch may further improve the V-gene inference. Indeed, a recent report by Ward and colleagues demonstrates that ModelAngelo and its *HMMer* search could be implemented to correlate high-resolution EMPEM data with B-cell receptor mRNA sequencing data to speed up and improve success rates of antibody discovery compared to previously reported data analysis pipelines (*Ferguson et al., 2024*). In addition, from the benchmark dataset analyzed here we might learn what sequencing mistakes are common in ModelAngelo data. This could be used to adjust the conventional Smith-Waterman Alignment in Stitch accordingly, analogous to recent improvements in the alignment algorithm we implemented for MS data (*Schulte and Snijder, 2024*). Finally, the template matching step in Stitch is now solely based on the sequence of the ModelAngelo models, but we may expand this to a structure-based alignment to better place the error-prone sequence reads in the correct framework of the Ig-domains.

Next steps in our efforts are to bring together the EMPEM work and MS-based polyclonal antibody sequencing on antigen-Fab complexes purified from patient serum. The goal is to reconstruct functional monoclonal antibodies from these analyses as ultimate proof for the accuracy of the derived antibody sequences and then start working on the throughput and depth of coverage in the repertoire. We believe that both EMPEM and MS-based polyclonal antibody sequencing are still limited to the top 1–10 antibodies in the polyclonal mixture. The EMPEM approach is biased towards bigger and well-ordered target antigens, which calls for additional complementary approaches like HDX-MS for a comprehensive polyclonal epitope mapping exercise. Bringing together these perspectives in an integrated structural biology approach promises new insights into the serum compartment of the antibody repertoire to better understand the coevolutionary processes of antibody maturation and antigenic drift.

## Materials and methods

### Benchmark of monoclonal antibody-antigen pairs and EMPEM maps from EMDB

We searched the EMDB for published maps containing antigen-binding fragments (Fabs) of any species, at nominal resolutions of ≤4 Ångström, released after the training data for ModelAngelo was obtained (April 1st 2022). The search was performed on February 11th 2023 at https://www.ebi. ac.uk/emdb/ using the search term 'antibody fab resolution:[* TO 4] AND release_date:[2022-03-31T00:00:00Z TO 2023-11-02T00:00:00Z] sample_name:*fab* fitted_pdbs:[* TO *]'. These results were filtered for maps that included a deposited atomic model and which contained only a single unique monoclonal Fab (of which multiple copies may be bound in the reconstructed antigen complex). When multiple redundant maps from the same study were included, we selected the single representative map of the highest quality, based on manual inspection. Global FSC resolution was not a good indicator as the maps were often dominated by the bound antigen, which may be better resolved than the bound antibody. Typically, the selected map was the focused/local refinement around the epitope-paratope region, despite its lower nominal resolution. A full overview of the selected maps

is provided in *Supplementary file 1*. The EMPEM maps were compiled based on a literature survey, complemented with a search of the EMDB using the term 'polyclonal'. A full overview of the selected EMPEM maps is provided in *Supplementary file 2*.

## Changes made to Stitch to take CIF input

Stitch was extended to allow mmCIF files as input. From these files all chains were extracted as separate amino acid sequences to align in Stitch. ModelAngelo outputs a confidence score per residue in the B-factor column of the mmCIF input, which was used as a local confidence for the sequence. First, each polypeptide gets assigned an Average Local Confidence (ALC) score based on the average across all residues, which can be used as an input filter on the data (along with polypeptide length). Second, the local confidence is used as weight in determining the consensus sequence of overlapping polypeptides, following assembly in Stitch.

## Analysis of monoclonal antigen-antibody and EMPEM benchmarks

Software used in this project was curated by SBGrid (*Morin et al., 2013*). The deposited model mmCIF and EM maps for the full benchmark were downloaded and ran with ModelAngelo (version 1.01) in 'build_no_seq' mode. For each entry in the benchmark the deposited mmCIF was run with Stitch (version 1.5.0-rc.1+6d3b540) using CutoffALC 80, minimum length 5 and TemplateMatching CutoffScore 8. The same was done for each mmCIF file produced by ModelAngelo but with an additional segment containing the antigen and Ig constant domain template sequences. From these runs, the consensus sequence and highest scoring germline for IGHV and IGLV (lambda +kappa) were retrieved. For each consensus sequence, the CDR3 was determined if the flanking cysteine on the V gene and the tryptophan or phenylalanine on the J gene were present. As these conserved residues were not all positioned correctly in the IMGT database, the data was manually fixed based on the same rules. The data from the deposited and produced Stitch runs was compared to produce the identity between the consensus sequences, distance between the inferred germlines, and identity between the CDR3s. The script used for this analysis is deposited as Supplementary Data. The results generated by this analysis is included in *Supplementary file 1*. The EMPEM benchmark was downloaded, ran through ModelAngelo and subsequently Stitch with identical parameters as above. In contrast to the benchmark detailed before, the ground-truth sequences of these antibodies is not known.

## Analysis of CR3022 data

The EM data for CR3022 was downloaded from EMDB (EMD-11648) and run with ModelAngelo (version 1.01) in 'build_no_seq' mode. The raw data for monoclonal antibodies CR3022, 107, 1028, 2771, 3576, and 3634 from PRIDE PXD030094 was downloaded and analyzed with PEAKS 10+. These listed monoclonal antibodies were chosen because these are the five most distant sequences from CR3022 and therefore present the biggest challenge to sequence assembly in Stitch (version 1.5.0-rc.1+6d3b540). The PEAKS analysis for the COVID-19 data from PRIDE PXD031941 was downloaded. Three sets of input were prepared: 'no background' consisting of only the CR3022 data, '+whole IgG' consisting of the CR3022 and the COVID-19 data, and '+5 mAbs' which consists of all mAbs from the CR3022 study. Three Stitch configurations where prepared: 'True' using the known CR3022 sequence as template as retrieved from PDB 7A5R, 'IMGT' using the conventional configuration of Stitch with all IMGT germlines as templates, and 'MA' using the closest V gene germline to the ModelAngelo consensus sequence of CR3022 together with the CDR3 sequence present. Each of these configurations were run with Stitch Recombine Decoy off and on, with this on for 'True' and 'MA' the full IMGT germline database was added and for 'IMGT' the Stitch parameter Decoy in Recombine was set allowing any unused germline from the Template Matching step to matched in the Recombine step. For all Stitch runs, the CutoffALC was 90 and the TemplateMatching CutoffScore 10. The resulting consensus sequences from these 18 Stitch runs were then compared with the known CR3022 sequence to determine the identity. The full script used for this analysis is included in the deposited Supplementary Data.

## Acknowledgements

This research was funded by the Dutch Research Council NWO Gravitation 2013 BOO, Institute for Chemical Immunology (ICI; 024.002.009), and the European Research Council Executive Agency HORIZON ERC-2022-STG (FLAVIR; 101077640).

## Additional information

### Funding

| Funder | Grant reference number | Author |
|---|---|---|
| Nederlandse Organisatie voor Wetenschappelijk Onderzoek | Gravitation 2013 BOO,Institute for Chemical Immunology (ICI; 024.002.009) | Douwe Schulte Marta Šiborová Joost Snijder |
| European Research Council | 10.13039/100019180 | Douwe Schulte Joost Snijder |

The funders had no role in study design, data collection and interpretation, or the decision to submit the work for publication.

### Author contributions

Douwe Schulte, Conceptualization, Data curation, Software, Formal analysis, Investigation, Visualization, Methodology, Writing - original draft, Writing – review and editing; Marta Šiborová, Data curation, Investigation, Writing – review and editing; Lukas Käll, Conceptualization, Investigation, Writing – review and editing; Joost Snijder, Conceptualization, Data curation, Formal analysis, Supervision, Funding acquisition, Investigation, Visualization, Methodology, Writing - original draft, Project administration, Writing – review and editing

### Author ORCIDs

Joost Snijder (iD) https://orcid.org/0000-0002-9310-8226

Reviewer #1 (Public review): https://doi.org/10.7554/eLife.101322.3.sa1
Reviewer #2 (Public review): https://doi.org/10.7554/eLife.101322.3.sa2
Author response https://doi.org/10.7554/eLife.101322.3.sa3

## Additional files

### Supplementary files

Supplementary file 1. Benchmark of experimental cryoEM maps of monoclonal antibody-antigen complexes from EMDB. Overview and results of maps for benchmark. *Name*: PDB-ID of deposited model for corresponding map *Actual*: corresponding to true VH/VL sequence as deposited in PDB model *Built*: corresponding to de novo sequences determined by ModelAngelo *Segment*: inferred V-gene from top-scoring hit in Stitch alignment *Score*: alignment score in Stitch *Identity*: sequence identity between *Actual-Built* based on de novo seq. or inferred *Segments Resolution*: Global FSC resolution of corresponding map *HC*: heavy chain *LC*: light chain

Supplementary file 2. Benchmark of EMPEM maps downloaded from EMDB for de novo modelling in ModelAngelo, with alignment scores in Stitch. Overview and results of maps for EMPEM benchmark.

MDAR checklist

### Data availability

Stitch is available at https://github.com/snijderlab/stitch (copy archived at *Schulte et al., 2024*). The CR3022 and COVID-19 whole IgG LC-MS/MS data were taken from PRIDE Archive (https://www.ebi.ac.uk/pride/archive/) via the PRIDE partner repository with the data set identifiers PXD030094 and PXD031941, respectively. All ModelAngelo and Stitch results, including a script to reproduce the full analysis, are made available on Zenodo under https://zenodo.org/records/12207014.

The following dataset was generated:

| Author(s) | Year | Dataset title | Dataset URL | Database and Identifier |
|---|---|---|---|---|
| Schulte D, Šiborová M, Käll L, Snijder J | 2024 | Supplementary data - Simultaneous polyclonal antibody sequencing and epitope mapping by cryo electron microscopy and mass spectrometry - a perspective | https://doi.org/ 10.5281/zenodo. 12207014 | Zenodo, 10.5281/ zenodo.12207014 |

The following previously published datasets were used:

| Author(s) | Year | Dataset title | Dataset URL | Database and Identifier |
|---|---|---|---|---|
| Faull P, Person M | 2022 | De Novo Sequencing of SARS-CoV-2 and influenza monoclonal antibodies by mass spectrometry using HCD and EThcD fragmentation and Supernovo software | https://www.ebi.ac. uk/pride/archive/ projects/PXD030094 | PRIDE, PXD030094 |
| Snijder J | 2022 | Template-based assembly of proteomic short reads for de novo antibody sequencing and repertoire profiling | https://www.ebi.ac. uk/pride/archive/ projects/PXD031941 | PRIDE, PXD031941 |

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
