## [Editor Report · eLife Assessment]

The paper addresses the problem of optimising the mapping of serum antibody responses against a known antigen. The manuscript describes a method using EM polyclonal epitope mapping to help elucidate endogenous antibodies. The work is interesting and **valuable** to the fields of immunology and serology, and the strength of evidence to support its findings is considered **solid**.

---

## [Referee Report · Reviewer #1 (Public review)]

Summary:

The paper addresses the problem of optimising the mapping of serum antibody responses against a known antigen. It uses the croEM analysis of polyclonal Fabs to antibody genes, with the ultimate aim of getting complete and accurate antibody sequences. The method, commonly termed EMPEM, is becoming increasingly used to understand responses in convalescent sera and optimisation of the workflows and provision of openly available tools is of genuine value to a growing number of people.

The authors do not address the experimental aspects of the methods and do not present novel computational tools, rather they use a series of established computational methods to provide workflows that simplify the interpretation of the EM map in terms of the sequences of dominant antibodies.

Strengths:

The paper is well-written and clearly argued. The tests constructed seem appropriate and fair and demonstrate that the workflow works pretty well. For a small subset (~17%) of the EMPEM maps analysed the workflow was able to get convincing assignments of the V-genes.

---

## [Referee Report · Reviewer #2 (Public review)]

In this manuscript, the authors seek to demonstrate that it is possible to sequence antibody variable domains from cryoEM reconstructions in combination with bottom-up LC-MSMS. In particular, they extract de novo sequences from single particle-cryo-EM-derived maps of antibodies using the "deep-learning tool ModelAngelo", which are run through the program Stitch to try to select the top scoring V-gene and construct a placeholder sequence for the CDR3 of both the heavy and light chain of the antibody under investigation. These reconstructed variable domains are then used as templates to guide the assembly of de novo peptides from LC-MS/MS data to improve the accuracy of the candidate sequence.

Using this approach the authors claim to have demonstrated that "cryoEM reconstructions of monoclonal antigen-antibody complexes may contain sufficient information to accurately narrow down candidate V-genes and that this can be integrated with proteomics data to improve the accuracy of candidate sequences".

---

## [Author Response]

The following is the authors’ response to the previous reviews

**Public Reviews:**

**Reviewer #1 (Public review):**
Summary:The paper addresses the problem of optimising the mapping of serum antibody responses against a known antigen. It uses the croEM analysis of polyclonal Fabs to antibody genes, with the ultimate aim of getting complete and accurate antibody sequences. The method, commonly termed EMPEM, is becoming increasingly used to understand responses in convalescent sera and optimisation of the workflows andThe authors do not address the experimental aspects of the methods and do not present novel computational tools, rather they use a series of established computational methods to provide workflows that simplify the interpretation of the EM map in terms of the sequences of dominant antibodies.

We would like to thank the reviewer for this assessment. While indeed we implement ModelAngelo as published without changes to its algorithms or code, we did add new functionality to Stitch to read the generated output from ModelAngelo and assemble it against known databases of germline-encoded antibody sequences. Of note, ModelAngelo was not primarily developed to determine exact sequence from CryoEM images, but instead to provide input for sequence determination from sequence searches with profile HMMs. Such models are designed to handle ambiguous calls of residues at different positions of a protein sequence. We are of the opinion that one of the main contributions of our study is to finally benchmark the EMPEM approach against known sequences to build a framework for data quality requirements in the future. From our study in best-case scenario’s EM data alone will provide sequences at 80-90% accuracy. In other words, the sequences are riddled with errors and cannot be taken at face value without orthogonal sequencing data. We demonstrate that mass spectrometry data can fill this requirement and yield much improved accuracy of the sequences even against high backgrounds of unrelated antibody sequences. We are incredibly excited about the prospects and future developments for EMPEM and believe that its integration with orthogonal sequencing approaches like MS are critical moving forward. By developing this pipeline we hope to have taken steps in the right direction.

Strengths:The paper is well-written and clearly argued. The tests constructed seem appropriate and fair and demonstrate that the workflow works pretty well. For a small subset (~17%) of the EMPEM maps analysed the workflow was able to get convincing assignments of the V-genes.

Thanks for the kind assessment.

Weaknesses:The AI methods used are not a substitute for high quality data and at present very few of the results obtained from EMPEM will be of sufficient quality to robustly assign the sequence of the antibody. However, rather more are likely to be good enough, especially in combination with MS data, to provide a pretty good indication of the V-gene family.

We fully agree with the assessment of the reviewer, as this being a general limitation of the EMPEM field. If anything, we hope our benchmark study and developed pipeline to integrate with MS-based sequencing data have more clearly established the current limitations of the technique and the requirements/prospects for orthogonal sequencing data to fill the missing gaps.

**Reviewer #2 (Public review):**
In this manuscript, the authors seek to demonstrate that it is possible to sequence antibody variable domains from cryoEM reconstructions in combination with bottom-up LC-MSMS. In particular, they extract de novo sequences from single particle-cryo-EM-derived maps of antibodies using the "deep-learning tool ModelAngelo", which are run through the program Stitch to try to select the top scoring V-gene and construct a placeholder sequence for the CDR3 of both the heavy and light chain of the antibody under investigation. These reconstructed variable domains are then used as templates to guide the assembly of de novo peptides from LC-MS/MS data to improve the accuracy of the candidate sequence.Using this approach the authors claim to have demonstrated that "cryoEM reconstructions of monoclonal antigen-antibody complexes may contain sufficient information to accurately narrow down candidate V-genes and that this can be integrated with proteomics data to improve the accuracy of candidate sequences".WhiIe the approach is clearly a work in progress, the manuscript should made easier to understand for the general reader. Indeed, I had a hard time understanding the workflow until I got to Fig. 3. So re-ordering the figures, for example, may be helpful in this regard.It would be useful to provide additional concrete examples where the described workflow would assist in the elucidation of CDR3's, in cases where this isn't already known. (In the benchmark dataset from the Electron Microscopy Data Bank, all the antibodies and Fabs are presumably known, as is the case for the monoclonal antibody CR3022). I am having difficulty envisioning how one would prepare samples from actual plasma samples that would be appropriate for single particle cryo-EM and MS data on dominant antibodies of interest. In my experience, most of these samples tend to be quite complex mixtures. So additional discussion of this point would be helpful.

We would like to thank the reviewer for their kind and critical assessment of our work. We have adopted the suggestion to reorder the graphical material, such that the workflow schematic is now Figure 1 in the main text. We hope this will improve the readability.

Regarding the concrete examples where the workflow could aid in elucidating CDR3 sequences, we would like to refer to all published EMPEM studies and in particular those highlighted in Figure 6. We are also actively working to integrate EMPEM data with MS-based sequencing on novel samples, but those will be subject of later studies. We have added additional discussion regarding the experimental feasibility of the approach. We have highlighted several milestone results where functional antibodies were reconstructed from EMPEM and/or MS data. In the discussion we write:

“While sample complexity remains an important bottleneck, and questions remain about the dynamic range of the true serum antibody repertoire and the depth of coverage from these novel experimental approaches, several studies have recently reached the important milestone of reconstructing functional antibodies from direct measurements of the secreted serum components.” (see references in manuscript)

“We believe that both EMPEM and MS-based polyclonal antibody sequencing are still limited to the top 1-10 antibodies in the polyclonal mixture. The EMPEM approach is biased towards bigger and well-ordered target antigens, which calls for additional complementary approaches like HDX-MS for a comprehensive polyclonal epitope mapping exercise.”

**Recommendations for the authors:**

**Reviewer #1 (Recommendations for the authors):**
Line 172: I am surprised the heavy chain is not worse than the light chain

We have added the following sentence:

“The length of the complete antigen binding loops was estimated with an average error of 0.5 ± 3.3 or 1.7 ± 6.0 residues for heavy and light chain, with average sequence identities of 0.63 and 0.41. While CDRH3 is the more challenging region in MS-based approaches to antibody sequencing, we believe that the moderately better length and sequence accuracy of CDRH3 compared to CDRL3 in ModelAngelo output reflects the CDRH3’s notoriously tight involvement in antigen binding, hence a greater relative stability in the antibody-antigen complex, resulting in better order in the reconstructed EM density maps.”

Line 175: Global FSC is not going to be useful. Why not use a local value?

We agree that local resolution estimates would be more appropriate, that is exactly why we added this remark to our initial analysis. However, local resolution estimates are non-trivial and raise the question about ‘how local’ we need to estimate the quality of the map (see for instance https://doi.org/10.1016/j.sbi.2020.06.005). At present, we believe that the required work for this local resolution analysis is not warranted, only to arrive at the rather intuitive if not tautological conclusion that a better map quality translates into more accurate sequences. While we agree that a better quantitative understanding of the data requirements for EMPEM could benefit the field, we opted to leave this, especially considering that the Stitch alignment score is already a good alternative predictor of sequence accuracy compared to map resolution as demonstrated in Figure 3,

Line 259: 'of the 23 maps' .... Actually there were 46 maps originally, so I feel this is a tad misleading.

The statistic of ‘46 total’ was added to the text.